# POLICY TRANSFER ENSURES FAST LEARNING FOR CONTINUOUS-TIME LQR WITH ENTROPY REGULARIZATION

## ABSTRACT

Reinforcement Learning (RL) enables agents to learn optimal decision-making strategies through interaction with an environment, yet training from scratch on complex tasks can be highly inefficient. Transfer learning (TL), widely successful in large language models (LLMs), offers a promising direction for enhancing RL efficiency by leveraging pre-trained models.

This paper investigates policy transfer, a TL approach that initializes learning in a target RL task using a policy from a related source task, in the context of continuous-time linear quadratic regulators (LQRs) with entropy regularization. We provide the first theoretical proof of policy transfer for continuous-time RL, proving that a policy optimal for one LQR serves as a near-optimal initialization for closely related LQRs, while preserving the original algorithm's convergence rate. Furthermore, we introduce a novel policy learning algorithm for continuous-time LQRs that achieves global linear and local super-linear convergence. Our results demonstrate both theoretical guarantees and algorithmic benefits of transfer learning in continuous-time RL, addressing a gap in existing literature and extending prior work from discrete to continuous time settings.

As a byproduct of our analysis, we derive the stability of a class of continuous-time score-based diffusion models via their connection with LQRs.

## 1 INTRODUCTION

**Transfer learning.** Transfer learning is a machine learning technique that leverages expertise gained from one domain to enhance the learning process in another related task. It is one of the most influential techniques that underpin the capabilities of large language models (LLMs). In the context of LLMs, transfer learning involves using pre-trained models, such as those from the GPT, BERT, or similar families, that were initially trained for specific tasks. Transfer learning repurposes these models for new and related applications, often involving domain-specific variations of the original problems. See *e.g.* Howard & Ruder (2018), Devlin et al. (2019), Raffel et al. (2020), Brown et al. (2020), Liu et al. (2019). Beyond LLMs, transfer learning has also gained a significant traction in other domains, particularly for improving learning efficiency when data and computational resources are limited. See *e.g.* Kraus & Feuerriegel (2017), Amodei et al. (2016), Tang et al. (2022).

**Reinforcement learning and transfer learning.** Reinforcement learning (RL) is one of the fundamental machine learning paradigms, where an agent learns to make a sequence of decisions by interacting with an environment and possibly with other agents. In a typical RL setup, an agent learns a policy/strategy for choosing actions in a given system state through trial and error to maximize a cumulative reward over time. However, training an agent for a complex RL task from the ground up can be extremely inefficient.

Given the exponentially growing demand for complex RL tasks, and the increasing number of pre-trained RL models for various learning tasks, it is natural to incorporate TL into RL to leverage knowledge from a pre-trained RL model to reduce both training time and computational costs, especially when there is a limited amount of data for new RL models.

Policy transfer is one of the most direct methods to leverage knowledge from one RL task to another. The basic idea of policy transfer is to use the policy learned from the source task to initialize the policy for the target task. If two RL tasks are similar, exploring the pre-trained policy as a starting point hopefully allows the agent to begin with a near-optimal strategy, with subsequent minor adjustments. This is intuitively clear and simple, and has been analyzed in a discrete-time LQ framework by Guo et al. (2023). Their work, as the first known theoretical studies for incorporating transfer learning into reinforcement learning, demonstrates the advantage in algorithmic performance improvement with TL technique for RL.

A natural question is, if the same benefit of transfer learning can be achieved for RL via appropriate policy transfer? Indeed, reinforcement learning, though primarily developed for discrete environment, is intrinsically continuous and complex, especially in robotics control, automatic driving, and portfolio optimization. However, analyzing transfer learning in the continuous-time RL framework remains uncharted and presents significantly greater technical challenges, as the knowledge to be transferred involves controlled stochastic processes and infinite-dimensional functional spaces.

**Our work.** This paper presents a theoretical analysis of policy transfer between continuous-time linear quadratic regulators with entropy regulation (LQRs). We demonstrate that an optimal policy learned for one LQR can serve as a near-optimal policy for any closely-related LQR, while preserving at least the same convergence rate as the original learning algorithm. In addition, we propose a novel policy learning algorithm for continuous-time LQRs, which achieves a global linear convergence rate and a local super-linear convergence rate. This implies that any closely related LQR is guaranteed with a super-linear convergent learning algorithm.

Our analysis fully exploits the Gaussian structure of the optimal policy for LQRs, as well as the robustness of the associated Riccati equation. As a byproduct of our analysis, we derive the stability of a class of continuous-time score-based diffusion models via their connection with LQRs.

**Related work.** The existing literature on policy learning for linear-quadratic (LQ) problems is extensive. For example, several studies focus on gradient-based algorithms for discrete-time LQRs. These algorithms, notably those proposed by Fazel et al. (2018) and Hambly et al. (2021), are able to achieve a global linear convergence rate in learning the parameters of the optimal feedback policy. Similarly, Giegrich et al. (2022) extends this approach to continuous-time LQRs, also demonstrating a global linear convergence rate. Beyond these gradient-based methods, other research explores different aspects of LQRs. For instance, Dean et al. (2020) develops a multistage procedure for designing a robust controller of discrete-time LQRs when the system dynamics are not fully known, while Huang et al. (2024) introduces a model-free algorithm that directly learns the optimal policy of continuous-time LQRs, providing a theoretical guarantee with a sublinear regret bound. Furthermore, Krauth et al. (2019) provides a theoretical analysis of the sample complexity of approximate policy iteration for learning discrete-time LQRs. For a more comprehensive background, interested readers are referred to the standard references by Kwakernaak & Sivan (1972) and Bertsekas (2019). Another closely related concept is the meta-learning for LQRs. For example, Toso et al. (2024) proves the first stability and convergence results of the *model-agnostic meta-learning* (MAML) in *discrete-time* LQRs.

The TL between MDPs is also well studied. For instance, Fu et al. (2023) investigates model transfer and policy transfer between *hidden-parameter MDPs* (HiP-MDPs), bounding the performance loss incurred by TL with the error in the estimation of hidden parameters. Lazaric & Restelli (2011) proposes sample-transfer algorithms and conducts the corresponding finite-sample analysis. Asadi et al. (2018) proves that, within the class of Lipschitz continuous MDPs, small perturbations in the dynamics only lead to a small change in the value function.

Our work of RL with TL is in *continuous time and state spaces*. The closest to our work is Guo et al. (2023), where a super-linear local convergent algorithm called IPO is proposed for *discrete-time* exploratory LQRs. In comparison, the analysis of policy transfer between continuous-time LQRs is technically more challenging and is on infinite-dimensional functional spaces. More importantly, we establish general results on policy transfer between any two closely related LQRs. The particular algorithm of IPO illustrates the benefit of policy transfer in such a context.

On the stability of continuous-time score-based diffusion models, Tang & Zhao (2024) has obtained a fairly general result under appropriate technical assumptions. Here we derive, via connecting

score-based diffusion models with LQRs, a class of models where these assumptions and hence the stability results hold.

Finally, the connection between score-based diffusion models and LQRs is well known. For example, Zhang & Katsoulakis (2023) shows that a large class of generative models, including normalizing flows, score-based diffusion models, and Wasserstein gradient flows, can be viewed as the solutions to certain mean-field games (MFGs). We note that LQRs can be viewed as the degenerate case of LQMFGs. Moreover, Gu et al. (2024) and Zhang et al. (2024) research the relationship between MFGs, Wasserstein proximals and score-based diffusion models.

**Notations.** For any smooth function $f : \mathbb{R}^n \to \mathbb{R}$, we use $\nabla f(x) \in \mathbb{R}^n$ to denote its gradient, and $\Delta f(x) \in \mathbb{R}^{n \times n}$ to denote its Hessian matrix. In addition, we use $\cdot$ to indicate the usual vector-vector and matrix-matrix inner products, depending on the context, and we use $S_{\geq 0}^n$ (*resp.* $S_{>0}^n$) to denote the space of $n \times n$ real positive semi-definite (*resp.* positive definite) matrices.

## 2 MATHEMATICAL FORMULATION

Let us now set up the mathematical framework under which entropy-regularized continuous-time linear quadratic regulators (LQRs) are defined over a finite time interval $[0, T]$.

Specifically, following the setup of Wang et al. (2018), we assume that the state process $x_t \in \mathbb{R}^n$ of the agent follows the linear SDE:

$$\mathrm{d}x_t = \Big[A_t x_t + B_t \mathbb{E}(u_t \,|\, x_t)\Big] \mathrm{d}t + \sigma_t \mathrm{d}W_t, \quad x_0 \sim \mathcal{D}_0, \tag{1}$$

where $\mathbb{E}[u_t \,|\, x_t] \sim \pi_t(\cdot \,|\, x_t) \in \mathcal{P}(\mathbb{R}^k)$ represents the randomized policy of the agent conditioned on $x_t$, $(W_t)_{t \in [0,T]}$ denotes the $d$-dimensional standard Brownian motion ($d$-BM for short), $\mathcal{D}_0$ denotes the initial distribution, and $(A_t, B_t, \sigma_t)_{t \in [0,T]}$ are appropriate deterministic matrix-valued processes to be specified later.

The agent minimizes the following entropy-regularized cost function:

$$\inf_{\pi \in \mathcal{A}} J_\pi(0, \mathcal{D}_0)$$

$$:= \mathbb{E}_{u_t \sim \pi_t(\cdot \,|\, x_t)} \left[ \int_0^T x_t^\dagger Q_t x_t + u_t^\dagger R_t u_t + \tau \log h_t(u_t \,|\, x_t) \mathrm{d}t + x_T^\dagger Q' x_T \,\middle|\, x_0 \sim \mathcal{D}_0 \right], \tag{2}$$

where $^\dagger$ denotes the transpose operator, $\mathcal{A}$ denotes the set of admissible randomized policies, $h_t(\cdot \,|\, x_t)$ denotes the conditional probability distribution function of $\pi_t(\cdot \,|\, x_t)$, and $(Q_t, R_t)_{t \in [0,T]}$ (*resp.* $Q'$) are appropriate deterministic matrix-valued processes (*resp.* matrix) to be specified later.

Note that the exploratory SDEs adopted here are first proposed by Wang et al. (2018), where an entropy-regularization term is added to the cost function to encourage agent exploration.

Next, we present the technical assumptions to ensure that the above formulation (1) – (2) is well defined. In particular, our goal is to ensure that (1) admits a unique strong solution (see *e.g.* (Oksendal, 2013, Theorem 5.2.1)) and that (2) has a finite integrand. See (Guo et al., 2022, Section 2) for a similar setup.

**Assumption 1** (Probability space). *We assume a filtered probability space* $(\Omega, \mathcal{F}, \mathbb{P}; (\mathcal{F}_t)_{t \in [0,T]})$ *where the filtration* $(\mathcal{F}_t)_{t \in [0,T]}$ *1) is rich enough to support some $d$-BM* $(W_t)_{t \in [0,T]}$, *the random action* $(u_t)_{t \in [0,T]}$ *of the agent, and the initial distribution* $\mathcal{D}_0$, *which are assumed to be independent; 2) satisfies the usual conditions (i.e., $\mathcal{F}_0$ contains all the $\mathbb{P}$-null sets and $(\mathcal{F}_t)_{t \in [0,T]}$ is right-continuous).*

**Assumption 2** (Admissible policies). *The set $\mathcal{A}$ of admissible policies consists of Markovian randomized policies,* i.e., *the following conditions hold for any $\pi \in \mathcal{A}$:*

  *1) for any $t \in [0,T]$ and $x \in \mathbb{R}^n$, $\pi_t(\cdot \,|\, x)$ is absolutely continuous w.r.t. the Lebesgue measure on $\mathbb{R}^k$ and has a finite expectation and a finite entropy;*

  *2) $\mathbb{E}(u_t \,|\, x)$, when viewed as a function of $(t, x) \in [0,T] \times \mathbb{R}^n$, has a linear growth w.r.t. $x$ and is Lipchitz continuous in $x$.*

**Assumption 3** (Regularity conditions). *$\mathcal{D}_0$ is assumed to be square integrable, and*

$$A, Q \in L^\infty([0, T], \mathbb{R}^{n \times n}), \quad B \in L^\infty([0, T], \mathbb{R}^{n \times k}),$$
$$R \in L^\infty([0, T], \mathbb{R}^{k \times k}), \quad \sigma \in L^2([0, T], \mathbb{R}^{n \times d}).$$

*In addition, we assume $Q_t \succeq 0$ a.e. for $t \in [0, T]$, $\tau > 0$, $Q' \succeq 0$, and that there exists $\delta > 0$ such that $R_t - \delta I \succeq 0$ a.e. for $t \in [0, T]$.*

## 3 TRANSFER LEARNING BETWEEN LQRs

In this section, we consider transfer learning between continuous-time linear quadratic regulators with entropy regulation (LQRs).

More specifically, suppose that there are two LQRs whose system parameters are $(\theta_t)_{t \in [0,T]}$ and $(\tilde{\theta}_t)_{t \in [0,T]}$, respectively. Without loss of generality, let us assume that the first LQR is more accessible and easier to learn, and let us denote by $(K_t^*)_{t \in [0,T]}$ the parameter of its optimal policy. We will show that if $(\theta_t)_{t \in [0,T]}$ and $(\tilde{\theta}_t)_{t \in [0,T]}$ are sufficiently close, then $(K_t^*)_{t \in [0,T]}$ may be used as an initialization to efficiently learn the optimal policy of the second LQR. Here in our framework the parameters $\theta = (A, Q, B, R, Q')$.

**Theorem 1** (Transfer learning of LQRs). *Given an LQR represented by model parameters $\theta$, for which there exists an optimal policy $\pi^*$ and an associated learning algorithm. Now, suppose there is a new LQR represented by model parameters $\tilde{\theta}$. Then, there exists $\epsilon > 0$, such that with an appropriate initialization $\pi^{(0)}$, this learning algorithm has the same convergence rate and finds a near-optimal policy of the new LQR, provided that*

$$d(\pi^{(0)}, \pi^*) + d(\tilde{\theta}, \theta) < \epsilon.$$

*Here $d$ denotes an appropriately chosen distance on the corresponding metric space.*

This result is based on the following two lemmas.

First, we see that the optimal randomized policy of the LQR defined by (1) – (2) can be derived via the dynamic programming principle (DPP) and by following a similar calculation from the earlier work Wang et al. (2018) and Guo et al. (2022).

**Lemma 2.** *The optimal randomized policy of the LQR (1) – (2) is:*

$$\pi_t^*(\cdot \,|\, x) = \mathcal{N}\left(-R_t^{-1} B_t^\dagger P_t x, \frac{\tau}{2} R_t^{-1}\right), \tag{3}$$

*where $P_t$ solves the following Riccati equation:*

$$\frac{\mathrm{d}P_t}{\mathrm{d}t} + A_t^\dagger P_t + P_t A_t + Q_t - P_t B_t R_t^{-1} B_t^\dagger P_t = 0, \quad P_T = Q'. \tag{4}$$

**Remark 1.** *The Gaussian form of $\pi^*$ originates from the entropy-regularization term in the cost function (2). The mean of $\pi^*$ appears in a mean-reverting fashion, pushing the agent to 0. Meanwhile, the covariance of $\pi^*$ is driven by the regularization coefficient $\tau > 0$. The larger the value of $\tau$, the more the agent would explore. In the case where $\tau \to 0^+$, $\pi^*$ would converge to a deterministic policy as one should expect (see (Wang et al., 2018, Section 5.4) for a formal discussion on the convergence of $\pi^*$).*

**Lemma 3** (Key lemma). *Under Assumption 3, denote by $\mathcal{R}$ the solution map of the Riccati equation (4), i.e.,*

$$\mathcal{R} : L^\infty([0, T], \mathbb{R}^{n \times n}) \times L^\infty([0, T], S_{\geq 0}^n) \times L^\infty([0, T], \mathbb{R}^{n \times k}) \times L^\infty([0, T], S_{>0}^k) \times S_{\geq 0}^n$$
$$\longrightarrow C([0, T], S_{\geq 0}^n)$$
$$(A_{t \in [0,T]}, Q_{t \in [0,T]}, B_{t \in [0,T]}, R_{t \in [0,T]}, Q') \longmapsto \mathcal{R}(A, Q, B, R, Q') := (P_t)_{t \in [0,T]}.$$

*Then, $\mathcal{R}$ is continuous, where the $L^\infty$ (resp. $S_{\geq 0}^n$) space is equipped with the functional $||\cdot||_{\infty;[0,T]}$ norm (resp. matrix 2-norm).*

Now, Theorem 1 follows immediately from Lemma 3. Indeed, by Lemma 3, the optimal policy is a continuous function in the LQR's model parameters. As a result, when the distance between $\tilde{\theta}$ and $\theta$ is small enough, the optimal policies of the two LQRs can be made arbitrarily close to each other. This implies the desired near-optimality.

## 4 IPO AND ITS SUPER-LINEAR CONVERGENCE FOR LQRs

Now we design an Iterative Policy Optimization (IPO) learning algorithm for LQRs. We will first establish its global linear convergence, and then show its super-linear convergence when the initial policy lies in a certain neighborhood of the optimal policy. As a corollary, in the context of transfer learning, we will see that such an algorithm yields an optimal policy for any closely related LQR with an appropriate initialization (*i.e.*, policy transfer). Our algorithm is analogous to the IPO algorithm developed for discrete-time LQRs in Guo et al. (2023), hence the adopted name IPO.

The algorithm and the analysis rely crucially on the Gaussian form of the LQR's optimal policy. Indeed, given the special form of (3), it suffices to optimize only within the following class of Gaussian policies:

$$\pi_t(\cdot \,|\, x) = \mathcal{N}(-K_t x, \Sigma_t), \tag{5}$$

where $K_t$ and $\Sigma_t$ are of appropriate shapes, and there exists $\delta_1 > 0$ such that $\Sigma_t - \delta_1 I \succeq 0$ for any $t \in [0, T]$. By (3), we observe that

$$K_t^* = R_t^{-1} B_t^\dagger P_t, \quad \Sigma_t^* = \frac{\tau}{2} R_t^{-1} \tag{6}$$

under the optimal policy of the LQR (1) – (2). First, we have

**DPP for the class of Gaussian policies.** Denote by $J^{K,\Sigma}$ the cost function associated with (5), with

$$J^{K,\Sigma}(t, x)$$

$$:= \mathbb{E}_{u_s \sim \pi_s(\cdot \,|\, x_s)} \left[ \int_t^T x_s^\dagger Q_s x_s + u_s^\dagger R_s u_s + \tau \log h_s(u_s \,|\, x_s) \mathrm{d}s + x_T^\dagger Q' x_T \,\Big|\, x_t = x \right].$$

Next, by DPP, $J^{K,\Sigma}$ satisfies the following Bellman equation:

$$\frac{\partial J^{K,\Sigma}}{\partial t} + \left[ (A_t - B_t K_t) x \right] \cdot \nabla J^{K,\Sigma} + \frac{1}{2} (\sigma_t \sigma_t^\dagger) \cdot \Delta J^{K,\Sigma}$$

$$+ x^\dagger (Q_t + K_t^\dagger R_t K_t) x + \mathrm{tr}(\Sigma_t R_t) - \frac{\tau}{2} \left[ k + \log \left( (2\pi)^k |\Sigma_t| \right) \right] = 0, \tag{7}$$

with the terminal condition $J^{K,\Sigma}(T, x) = x^\dagger Q' x$. By plugging in the ansatz

$$J^{K,\Sigma}(t, x) = x^\dagger P_t^K x + r_t^{K,\Sigma},$$

we obtain the coupled Riccati equations:

$$\frac{\mathrm{d} P_t^K}{\mathrm{d} t} + (A_t - B_t K_t)^\dagger P_t^K + P_t^K (A_t - B_t K_t) + Q_t + K_t^\dagger R_t K_t = 0, \quad P_T^K = Q', \tag{8}$$

$$\frac{\mathrm{d} r_t^{K,\Sigma}}{\mathrm{d} t} + \mathrm{tr}(\sigma_t^\dagger P_t^K \sigma_t + \Sigma_t R_t) - \frac{\tau}{2} \left[ k + \log \left( (2\pi)^k |\Sigma_t| \right) \right] = 0, \quad r_T^{K,\Sigma} = 0. \tag{9}$$

Note that $P_t^K$ only depends on $K_t$, and $r_t^{K,\Sigma}$ depends on $(K_t, \Sigma_t)$. Recall that Assumption 3 is sufficient for (8) to admit a unique $C^1$ solution taking values in $S_{\geq 0}^n$ (*cf.* (Yong & Zhou, 2012, Corollary 2.10)).

Now we can derive an IPO algorithm for updating the parameters in the Gaussian policy (5), namely $K_t$ and $\Sigma_t$, with the goal of learning the parameters of the optimal randomized policy, which are denoted by $K_t^*$ and $\Sigma_t^*$ (*cf.* (6)).

**Iterative policy optimization (IPO) derivation.** For any $\Delta t > 0$, $J^{K,\Sigma}(t, x)$ satisfies the Bellman equation:

$$J^{K,\Sigma}(t, x) = \mathbb{E}_{u \sim \pi_{K,\Sigma}} \left[ \int_t^{t+\Delta t} x_s^\dagger Q_s x_s + u_s^\dagger R_s u_s + \tau \log h_s(u_s \,|\, x_s) \mathrm{d}s \right.$$

$$\left. + J^{K,\Sigma}(t + \Delta t, x_{t+\Delta t}) \,\Big|\, x_t = x \right]. \tag{10}$$

We define the *preliminary IPO algorithm* of $(K_t, \Sigma_t)$ by:

$$K_t^{\text{prelim}}, \Sigma_t^{\text{prelim}}$$

$$:= \underset{\widetilde{K}, \widetilde{\Sigma}}{\operatorname{argmin}} \, \mathbb{E}_{u \sim \pi_{\widetilde{K}, \widetilde{\Sigma}}} \left[ \int_t^{t+\Delta t} x_s^\dagger Q_s x_s + u_s^\dagger R_s u_s + \tau \log h_s(u_s | x_s) \mathrm{d}s \right.$$

$$\left. + J^{K,\Sigma}(t + \Delta t, x_{t+\Delta t}) \,\Big|\, x_t = x \right],$$

which depends on the value of $\Delta t$ and is equivalent to:

$$K_t^{\text{prelim}}, \Sigma_t^{\text{prelim}}$$

$$:= \underset{\widetilde{K}, \widetilde{\Sigma}}{\operatorname{argmin}} \, \mathbb{E}_{u \sim \pi_{\widetilde{K}, \widetilde{\Sigma}}} \left[ \frac{1}{\Delta t} \int_t^{t+\Delta t} x_s^\dagger Q_s x_s + u_s^\dagger R_s u_s + \tau \log h_s(u_s \,|\, x_s) \mathrm{d}s \right.$$

$$\left. + \frac{1}{\Delta t} \left[ J^{K,\Sigma}(t + \Delta t, x_{t+\Delta t}) - J^{K,\Sigma}(t, x) \right] \Big|\, x_t = x \right]. \quad (11)$$

Our *IPO algorithm* is then defined by the limit of the above preliminary algorithm, that is, on the RHS of (11), we set $\Delta t \to 0^+$ and exchange the limit with $\operatorname{argmin}$ to obtain (*i.e.*, minimizing the first-order derivative of the RHS of (10) at $\Delta t = 0$):

$$K_t^{\text{IPO}}, \Sigma_t^{\text{IPO}} := \underset{\widetilde{K}_t, \widetilde{\Sigma}_t}{\operatorname{argmin}} \left\{ x^\dagger (\widetilde{K}_t^\dagger R_t \widetilde{K}_t - 2\widetilde{K}_t^\dagger B_t^\dagger P_t^K) x + \operatorname{tr}(\widetilde{\Sigma}_t R_t) - \frac{\tau}{2} \log |\widetilde{\Sigma}_t| \right\},$$

which admits the following analytical solution:

$$K_t^{\text{IPO}} = R_t^{-1} B_t^\dagger P_t^K, \quad \text{(IPO: } K\text{)}$$

$$\Sigma_t^{\text{IPO}} = \frac{\tau}{2} R_t^{-1}. \quad \text{(IPO: } \Sigma\text{)}$$

where $P_t^K$ is the solution to (8). Notice that $\Sigma_t^{\text{IPO}}$ reaches the covariance of the optimal Gaussian policy after a single iteration (*cf.* (6)). We present below the IPO algorithm for updating $K_t$.

---

**Algorithm 1** IPO algorithm for learning $(K_t^*)_{t \in [0,T]}$

---

**Require:** Initial value $(K_t^{(0)})_{t \in [0,T]}$
1: $i \leftarrow 0$
2: **while** not stop_flag **do**
3:      Solve (8) to obtain $(P_t^{K^{(i)}})_{t \in [0,T]}$
4:      $K_t^{(i+1)} \leftarrow R_t^{-1} B_t^\dagger P_t^{K^{(i)}}$, $t \in [0, T]$
5:      $i \leftarrow i + 1$
6: **end while**
7: **return** $(K_t^{(i)})_{t \in [0,T]}$

---

**Convergence of IPO.** Now we present the convergence results of the IPO algorithm defined by (IPO: $K$) – (IPO: $\Sigma$). We will show that with an additional assumption stated in Assumption 4, the IPO algorithm has a global linear convergence rate. Since $(\Sigma_t^{\text{IPO}})_{t \in [0,T]}$ always reaches the covariance of the optimal Gaussian policy after a single iteration, we only discuss the convergence of $(K_t^{\text{IPO}})_{t \in [0,T]}$.

For any given parameters $(K_t, \Sigma_t)_{t \in [0,T]}$, we use the cost function value to measure their goodness (with an abuse of notation):

$$C(K, \Sigma) := J_{\pi_{K,\Sigma}}(0, \mathcal{D}_0) \quad (12)$$

$$= \mathbb{E}\left( x^\dagger P_0^K x + r_0^{K,\Sigma} \,\Big|\, x \sim \mathcal{D}_0 \right),$$

where $(P_t^K, r_t^{K,\Sigma})_{t \in [0,T]}$ solves the coupled Riccati equations (8) – (9). Note that $C(K, \Sigma)$ is minimized at $(K_t^*, \Sigma_t^*)_{t \in [0,T]}$ (*resp.* at $(K_t^*)_{t \in [0,T]}$ when viewed only as a functional in $K$). See (6) for the values of $(K_t^*, \Sigma_t^*)_{t \in [0,T]}$.

**Assumption 4.** $\mathbb{E}\big(x_0 x_0^\dagger \mid x_0 \sim \mathcal{D}_0\big) \succ 0.$

**Theorem 4** (Global linear convergence of IPO). *Under Assumptions 1 – 4, suppose that* $\{(K_t^{(i)}, \Sigma_t)_{t \in [0,T]}\}_{i \geq 0}$ *is a sequence of parameters following the algorithm* (IPO: $K$). *Then, there exist constants* $\mathcal{C}_1^K > 0$ *and* $0 \leq \mathcal{C}_1 < 1$, *which depend on* $K^{(0)}$ *and the data of the LQR* (1) – (2), *such that:*

$$\forall i \geq 0, \quad \mathcal{C}_1^K \int_0^T \left\| K_t^{(i+1)} - K_t^* \right\|_2^2 \mathrm{d}t \leq C(K^{(i+1)}, \Sigma) - C(K^*, \Sigma)$$

$$\leq \mathcal{C}_1 \left[ C(K^{(i)}, \Sigma) - C(K^*, \Sigma) \right].$$

One can further establish a super-linear convergence rate for the IPO algorithm, with an appropriate initialization.

**Theorem 5** (Local super-linear convergence of IPO). *Under Assumptions 1 – 4, there exist constants* $(\epsilon, \mathcal{C}_2) > 0$, *which depend on the data of the LQR* (1) – (2), *such that for any sequence of parameters* $\{(K_t^{(i)}, \Sigma_t)_{t \in [0,T]}\}_{i \geq 0}$ *following the algorithm* (IPO: $K$) *and satisfying:*

$$\int_0^T \left\| K_t^{(0)} - K_t^* \right\|_2^2 \mathrm{d}t \leq \epsilon,$$

*the following local super-linear convergence holds:*

$$\forall i \geq 0, \quad C(K^{(i+1)}, \Sigma) - C(K^*, \Sigma) \leq \mathcal{C}_2 \left[ C(K^{(i)}, \Sigma) - C(K^*, \Sigma) \right]^{\frac{3}{2}}.$$

**Remark 2.** *Assumption 4 is critical in proving that the minimum eigenvalue of* $\mathbb{E}(x_t x_t^\dagger)$ *is uniformly bounded away from 0 (cf. Lemma 11). This uniform lower bound then leads to the uniform contraction of the IPO algorithm. In the discrete-time setting (cf. Guo et al. (2023)), the counterpart of Assumption 4 is also imposed to guarantee the global linear convergence of the algorithms (cf. (Guo et al., 2023, Lemma 5.2)).*

**Remark 3.** *In fact, one can replace* $\mathcal{D}_0$ *with any square-integrable distribution in the definition of* $C(\cdot, \cdot)$ *(cf. (12)) and all the above convergence results still hold. This is because the initial distribution of the LQR* (1) – (2) *is irrelevant to the definition of the IPO algorithm. In this case, one only needs to change the statement of Assumption 4 to guarantee the corresponding positive-definiteness.*

**Transfer learning with IPO.** Now combining Theorem 1 and Theorem 5, we have immediately the super-fast learning via appropriate policy transfer between LQRs. We mention that in Theorem 5, $\epsilon$ admits a lower bound which only depends on the norms of the LQR's model parameters.

**Corollary 6** (Transfer learning of LQRs with IPO). *Under Assumptions 1 – 4, denote by* $(K_t^*)_{t \in [0,T]}$ *the parameter of the optimal Gaussian policy of the LQR represented by* $(A_{t \in [0,T]}, Q_{t \in [0,T]}, B_{t \in [0,T]}, R_{t \in [0,T]}, Q')$. *Then, there exists* $\epsilon > 0$, *such that any initialization* $(K_t^{(0)})_{t \in [0,T]}$ *converges with a super-linear convergence rate to the optimal Gaussian policy of any LQR represented by* $(\tilde{A}_{t \in [0,T]}, \tilde{Q}_{t \in [0,T]}, \tilde{B}_{t \in [0,T]}, \tilde{R}_{t \in [0,T]}, \tilde{Q}')$, *provided that*

$$||K^{(0)} - K^*||_{2;[0,T]} + ||\tilde{A} - A||_{\infty;[0,T]} + ||\tilde{Q} - Q||_{\infty;[0,T]}$$

$$+ ||\tilde{B} - B||_{\infty;[0,T]} + ||\tilde{R} - R||_{\infty;[0,T]} + ||\tilde{Q}' - Q'||_2 < \epsilon,$$

*where* $|| \cdot ||_{\infty;[0,T]}$ *(resp.* $|| \cdot ||_{2;[0,T]}, || \cdot ||_2$*) denotes the functional* $L^\infty$ *norm (resp. functional* $L^2$ *norm, matrix 2-norm).*

## 5 APPLICATION: STABILITY OF SCORE-BASED DIFFUSION MODELS

In this section, we will show that our analysis of LQRs, especially the key Lemma 3 can be applied to obtain the stability of score-based diffusion models. The critical observation is that the probability density function of (a certain class of) score-based diffusion models can be found in the LQRs under

the optimal randomized policy. This allows us to consider a class of score matching functions and to bound the distance between the generated distribution and the target distribution.

In the rest of this section, we always impose the following assumption on the LQR (1) – (2).

**Assumption 5.** *We assume*

$$\text{tr}(A_t) = -\frac{\tau}{4} \log \frac{|R_t|}{(\tau\pi)^k}, \quad B_t R_t^{-1} B_t^\dagger = \sigma_t \sigma_t^\dagger, \quad Q_t = 0$$

*for any $t \in [0, T]$, and $Q' \succ 0$.*

**Mechanism of score-based diffusion models.** Score-based diffusion models have become the SOTA solution to various tasks in different areas. For completeness, we first recall their basic mechanism briefly. (See *e.g.* Tang & Zhao (2024) for a comprehensive review).

Suppose $p_0^{\text{data}}$ is the distribution that one aims to generate. Diffusion model starts by defining a forward SDE (*e.g.* an OU process) over $[0, T]$ with the initial distribution $p_0^{\text{data}}$. Denote by $s$ and $p_T^{\text{data}}$ the score function and the terminal distribution of the forward SDE, respectively. Then, in the sampling stage, a backward SDE, whose dynamics depend on $s$ and whose initial distribution is $p_T^{\text{data}}$, is simulated. (Figure 1 summarizes the basic mechanism of score-based diffusion models).

Theoretically, it can be shown that the terminal distribution of the backward SDE is equal to $p_0^{\text{data}}$. In practice, however, $s$ and $p_T^{\text{data}}$ are typically not accessible, and a score matching function $s_\beta$ and a noise distribution $p^{\text{noise}}$ are adopted as their approximations, respectively. Denote by $p_\approx^{\text{data}}$ the terminal distribution of the backward SDE (under $s_\beta$ and $p^{\text{noise}}$).

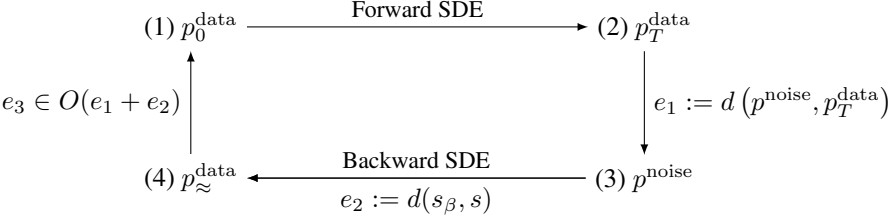

**Figure 1:** Basic mechanism of score-based diffusion models.

**Connection with LQRs.** Next, we recall the connection between score-based diffusion models and LQRs. Our Lemma 7 can be viewed as a special case of (Zhang & Katsoulakis, 2023, Theorem 7). The key ingredient is the Cole-Hopf transformation for the HJB equation that characterizes the optimal policy of the LQR (*cf.* (HJB) in our case).

**Lemma 7.** *Under Assumptions 1 – 3 and 5, the probability density function, which is denoted by $\hat{p}(t, x)$, of the following diffusion process on $[0, T]$:*

$$\mathrm{d}\hat{X}_t = -A_{T-t}\hat{X}_t \mathrm{d}t + \sigma_{T-t}\mathrm{d}W_t, \quad \hat{X}_0 \sim \mathcal{N}\left(0, (Q')^{-1}\right) \tag{13}$$

*can be expressed by:*

$$\hat{p}(t, x) = (2\pi)^{-\frac{n}{2}} |Q'|^{\frac{1}{2}} \exp\left[-\frac{1}{2} J(T-t, x)\right],$$

*where $J(t, x) = x^\dagger P_t x + r_t$ with $(P_t, r_t)$ solving the coupled Riccati equations (4) – (17).*

We note that (13) specifies a diffusion model where the data distribution $\hat{X}_0$ (*i.e.*, the distribution one aims to generate) is Gaussian, and the forward SDE is an OU process. By Lemma 7, $\hat{p}(t, x)$ is determined by $P_{T-t}$ and $r_{T-t}$. In fact, $\hat{p}(t, x)$ is determined solely by $P_{T-t}$ since the spacial integral of $\hat{p}(t, x)$ must be 1. As a result, the score function of (13) (*i.e.*, the gradient of $\log \hat{p}(t, x)$) is determined by $P_{T-t}$.

**A class of score matching functions and the stability.** Now, the backward SDE of (13) is:

$$\mathrm{d}\hat{Y}_t = \left[ A_t \hat{Y}_t + \sigma_t \sigma_t^\dagger \nabla \log \hat{p}^{Q'}(T - t, \hat{Y}_t) \right] \mathrm{d}t + \sigma_t \mathrm{d}W_t, \quad \hat{Y}_0 \sim \hat{p}^{Q'}(T, \cdot), \tag{14}$$

where we use $\hat{p}^{Q'}$ to indicate the dependence of $\hat{p}$ on $Q'$.

In practice, when $\hat{p}^{Q'}$ is not explicitly known, a score matching function $s$ is used as an approximator of $\nabla \log \hat{p}^{Q'}$, and the initial distribution is approximated by some noise distribution $p^{\mathrm{noise}}$, *i.e.*,

$$\mathrm{d}Y_t = \left[ A_t Y_t + \sigma_t \sigma_t^\dagger s(T - t, Y_t) \right] \mathrm{d}t + \sigma_t \mathrm{d}W_t, \quad Y_0 \sim p^{\mathrm{noise}}. \tag{15}$$

We will show that $Y_T \approx \hat{Y}_T \stackrel{d}{=} \hat{X}_0$ when $s \approx \nabla \log \hat{p}^{Q'}$ and $p^{\mathrm{noise}} \approx \hat{p}^{Q'}(T, \cdot)$: this follows from the stability of the Riccati equation (4) (*cf.* Lemma 3), such that $s = \nabla \log \hat{p}^{M}$ serves as a good score matching function as long as $M \approx Q'$.

**Theorem 8** (Error bound analysis). *Under Assumptions 1 − 3 and 5, there exist constants $(C_1, C_2, C_3) > 0$, which depend on the data of the LQR (1) – (2), such that for any $\epsilon > 0$, there exists $\delta_0 > 0$, such that $||M - Q'|| < \delta_0$ implies*

$$d_{\mathrm{TV}} \left( Y_T, \hat{Y}_T \right) \leq d_{\mathrm{TV}} \left( p^{\mathrm{noise}}, \hat{p}^{Q'}(T, \cdot) \right) + C_1 \epsilon,$$

*and*

$$W_2 \left( Y_T, \hat{Y}_T \right) \leq \sqrt{C_2 W_2^2 \left( p^{\mathrm{noise}}, \hat{p}^{Q'}(T, \cdot) \right) + C_3 \epsilon^2},$$

*where $Y_t$ satisfies (15) with $s = \nabla \log \hat{p}^{M}$, and $\hat{Y}_t$ satisfies (14). Here $d_{\mathrm{TV}}$ and $W_2$ to denote the total variation distance and 2-Wasserstein distance, respectively.*

*Proof.* Our proof utilizes the results in (Tang & Zhao, 2024, Section 5). We first prove the total variation bound. By Lemma 3, for any fixed $x \in \mathbb{R}^n$, we have:

$$\nabla \log q^M(\cdot, x) \to \nabla \log q^{Q'}(\cdot, x) \text{ in } C([0, T], \mathbb{R}^n)$$

as $M \to Q'$. Then, by probability theory, we have:

$$\forall t \in [0, T], \quad \mathbb{E}_{\hat{X}_t \sim q(t, \cdot)} \left|\left| \nabla \log q^M(t, \hat{X}_t) - \nabla \log q^{Q'}(t, \hat{X}_t) \right|\right|^2 \to 0$$

as $M \to Q'$. The total variation bound is then proved by invoking (Tang & Zhao, 2024, Theorem 5.2). Similarly, the 2-Wasserstein bound can be proved by invoking (Tang & Zhao, 2024, Theorem 5.5 and Eqn. (5.13)). □

## 6 NUMERICAL EXPERIMENTS

In this section, we conduct a toy numerical experiment to illustrate our main convergence results Theorem 4 (*i.e.*, global linear convergence) and Theorem 5 (*i.e.*, local super-linear convergence) of the IPO algorithm 1. We assume that the values of $(A_{t \in [0,T]}, B_{t \in [0,T]}, Q_{t \in [0,T]}, R_{t \in [0,T]}, Q')$ in (1) – (2) are known.

**Choice of model parameters.** For simplicity, we only consider the case where the matrix-valued processes $(A_{t \in [0,T]}, B_{t \in [0,T]}, Q_{t \in [0,T]}, R_{t \in [0,T]})$ are constant in $t$. We choose $T = 1$, $(n, k) = (3, 2)$, and the values of the model parameters are sampled independently from the standard normal distribution $\mathcal{N}(0, 1)$ [1]. At each time step $t$, the parameter $K_t^{(0)}$ of the initial policy is also sampled independently from $\mathcal{N}(0, 1)$. Note that we do not require $(K_t^{(0)})_{t \in [0,T]}$ to be constant in $t$.

---

[1]To sample a PSD matrix, we first sample a random matrix and then multiply it with its transpose.

**Numerical results.** The convergence of our IPO algorithm is plotted in Figure 2. The $x$-axis shows the iteration and the $y$-axis shows the mean $L^2$ error, where $(K_t^{(i)})_{t \in [0,T]}$ denotes the Gaussian policy's parameter at the $i$-th iteration, and $(K_t^*)_{t \in [0,T]}$ denotes the parameter of the optimal Gaussian policy. As clearly shown, the algorithm admits linear convergence at the early stages and then super-linear convergence when the policy approaches the optimum, which empirically verifies our theoretical results Theorem 4 (*i.e.*, global linear convergence) and Theorem 5 (*i.e.*, local super-linear convergence).

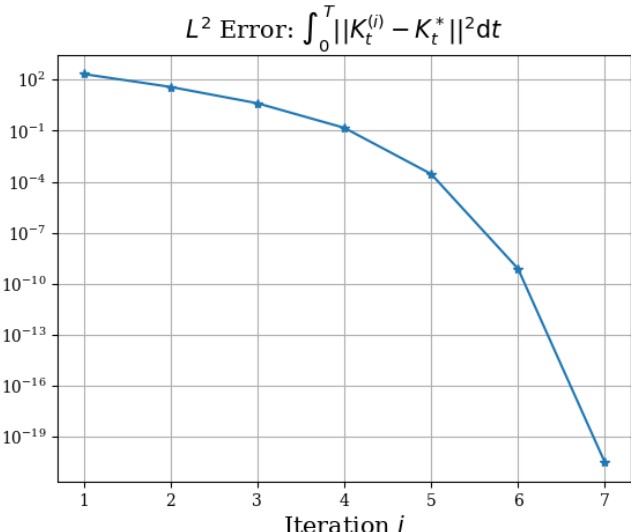

**Figure 2:** Convergence of the IPO Algorithm 1.

**Conclusion.** Linear-quadratic (LQ) control problems are a cornerstone of classical control theory. Our analysis of transfer learning for LQRs benefits from its analytical tractability and gains critical insights for general continuous-time RL problems. In particular, it shows that transfer learning will be valuable for leveraging existing RL algorithms beyond LQR framework. The precise mathematical analysis relies on studies of stability and continuity of optimal policy for stochastic control problems. The analysis on LQRs also leads to the stability results for a class of score-based continuous-time diffusion models.

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

# 7 APPENDIX

## 7.1 PROOF OF LEMMA 2

*Proof.* Define the following intermediate cost function:

$$J(t,x) := \inf_{\pi \in \mathcal{A}} \mathbb{E}_{u_s \sim \pi_s(\cdot \mid x_s)} \left[ \int_t^T x_s^\dagger Q_s x_s + u_s^\dagger R_s u_s + \tau \log h_s(u_s \mid x_s) \mathrm{d}s \right.$$

$$\left. + x_T^\dagger Q' x_T \,\middle|\, x_t = x \right]. \quad (16)$$

Then, DPP produces the following HJB equation of the LQR (1) – (2):

$$-\frac{\partial J(t,x)}{\partial t} = \inf_{\pi \in \mathcal{A}} \mathbb{E}_{u_t \sim \pi_t} \left\{ (A_t x + B_t \bar{u}_t) \cdot \nabla J(t,x) + \frac{1}{2}(\sigma_t \sigma_t^\dagger) \cdot \Delta J(t,x) \right.$$

$$\left. + x^\dagger Q_t x + u_t^\dagger R_t u_t + \tau \log h_t(u_t \mid x) \right\}, \quad J(T,x) = x^\dagger Q' x, \quad \text{(HJB)}$$

where we use $\bar{\cdot}$ to imply the expectation of the underlying random variable/distribution, and $h_t(\cdot \mid x)$ denotes the (conditional) probability distribution function of the Markov randomized policy $\pi_t(\cdot \mid x)$.

By (Guo et al., 2023, Lemma 2.2), the RHS of (HJB) is minimized by the following Gaussian policy:

$$\pi_t^*(\cdot \mid x) = \mathcal{N}\left( -\frac{1}{2} R_t^{-1} B_t^\dagger \nabla J(t,x), \frac{\tau}{2} R_t^{-1} \right).$$

Observing the linear quadratic nature of LQRs, we introduce the following ansatz for $J(t,x)$:

$$J(t,x) = x^\dagger P_t x + r_t.$$

After plugging the ansatz for $J$ and the expression of $\pi^*$ into (HJB), we obtain the following coupled Riccati equations for $(P_t, r_t)$:

$$\frac{\mathrm{d}P_t}{\mathrm{d}t} + A_t^\dagger P_t + P_t A_t + Q_t - P_t B_t R_t^{-1} B_t^\dagger P_t = 0, \quad P_T = Q', \quad (4)$$

$$\frac{\mathrm{d}r_t}{\mathrm{d}t} + \mathrm{tr}(\sigma_t^\dagger P_t \sigma_t) + \frac{\tau}{2} \log \frac{|R_t|}{(\tau\pi)^k} = 0, \quad r_T = 0. \quad (17)$$

Hence the lemma. □

## 7.2 APPENDIX B: PROOF OF LEMMA 3

*Proof.* The well-definedness of $\mathcal{R}$ is guaranteed by (Yong & Zhou, 2012, Corollary 2.10). For simplicity, in the rest we only prove the continuity of $\mathcal{R}$ with respect to $(A_t, B_t)_{t \in [0,T]}$. The continuity with respect to the rest arguments can be proved by the same route.

For any $(A_t, B_t)$ (*resp.* $(\tilde{A}_t, \tilde{B})$), we denote by $P_t$ (*resp.* $\tilde{P}_t$) the solution of (4). Define $\Delta P_t := P_t - \tilde{P}_t$. It can be shown that $\Delta P_t$ satisfies the ODE:

$$\frac{\mathrm{d}\Delta P_t}{\mathrm{d}t} + A_t^\dagger \Delta P_t + (A_t - \tilde{A}_t)^\dagger \tilde{P}_t + \Delta P_t A_t + \tilde{P}_t(A_t - \tilde{A}_t)$$

$$- (P_t B_t R_t^{-1} B_t^\dagger P_t - \tilde{P}_t \tilde{B}_t R_t^{-1} \tilde{B}_t^\dagger \tilde{P}_t) = 0, \quad \Delta P_T = 0.$$

By integrating over $[t,T]$ and then taking the matrix 2-norm on both sides, we obtain:

$$||\Delta P_t||_2 \le \int_t^T \left[ 2||A_s||_2 + \delta||B_s||_2^2(||P_s||_2 + ||\tilde{P}_s||_2) \right] ||\Delta P_s||_2 \mathrm{d}s$$

$$+ 2||A - \tilde{A}||_{\infty;[0,T]} \int_t^T ||\tilde{P}_s||_2 \mathrm{d}s$$

$$+ 2\delta||B - \tilde{B}||_{\infty;[0,T]} \int_t^T (||B_s||_2||P_s||_2 + ||\tilde{B}_s||_2||\tilde{P}_s||_2) ||\tilde{P}_s||_2 \mathrm{d}s, \quad (18)$$

where $\delta > 0$ is defined in Assumption 3. Notice that $||P||_{\infty;[0,T]}$ (*resp.* $||\tilde{P}||_{\infty;[0,T]}$) can be controlled by $||A||_{\infty;[0,T]} + ||B||_{\infty;[0,T]}$ (*resp.* $||\tilde{A}||_{\infty;[0,T]} + ||\tilde{B}||_{\infty;[0,T]}$), applying Gronwall's inequality on (18) finishes the proof. $\qquad\square$

### 7.3 PROOF OF THEOREM 4

In the following, with an abuse of notation, we sometimes use $\langle \cdot, \cdot \rangle$ to indicate the usual matrix inner product. In addition, for any matrix $M$, we use $\lambda_{\min}(M)$ (*resp.* $||M||_2$) to denote the square root of the smallest (*resp.* largest) eigenvalue of $M^\dagger M$.

We first define the following matrix-valued functions.

$$\mathcal{G}(t, K', K) := P_t^K \left[ B_t(K_t - K'_t) \right] + \left[ B_t(K_t - K'_t) \right]^\dagger P_t^K + K_t'^\dagger R_t K_t' - K_t^\dagger R_t K_t,$$

$$G(t, K) := -\mathcal{G}(t, R^{-1}B^\dagger P^K, K) = P_t^K B_t R_t^{-1} B_t^\dagger P_t^K + K_t^\dagger R_t K_t - P_t^K B_t K_t - K_t^\dagger B_t^\dagger P_t^K.$$

Since $K_t^* = R_t^{-1}B_t^\dagger P_t^*$, we have $\mathcal{G}(t, K, K^*) = (K_t - K_t^*)^\dagger R_t(K_t - K_t^*) \succeq 0$. Also, it can be verified by algebraic calculation that $G(t, K) \succeq 0$. In addition, for notational simplicity, in the rest of this section, we define

$$y_t := \mathbb{E}(x_t x_t^\dagger), \tag{✿}$$

where $x_t$ solves the state SDE (1) with $u_t$ following the policy $\pi_t = \mathcal{N}(-K_t x, \Sigma_t)$. And we shall use superscripts to indicate different policies. For instance, by $y'_t$ we imply that $y'_t = \mathbb{E}\left( x'_t(x'_t)^\dagger \right)$ where $x'_t$ solves the state SDE (1) with $u_t$ following the policy $\pi_t = \mathcal{N}(-K'_t x, \Sigma_t)$.

The proof of the global linear convergence relies on the following lemmas.

**Lemma 9** (Cost difference). *Under Assumptions 1 – 3, the cost difference of two parametrized Gaussian policies is given by:*

$$C(K', \Sigma) - C(K, \Sigma) = \int_0^T \left\langle y'_t, \mathcal{G}(t, K', K) \right\rangle \mathrm{d}t,$$

*where $y'_t$ is defined by (✿).*

*Proof.* Under Assumptions 1 – 3, recall the definition of associated cost function from Section 4. For notational simplicity, denote $J'(t, x) := J_{K',\Sigma}(t, x)$ and $J(t, x) := J^{K,\Sigma}(t, x)$. By subtracting the two Bellman equations that $J'(t, x)$ and $J(t, x)$ satisfy (*cf.* (7)), we obtain:

$$\frac{\partial(J' - J)}{\partial t} + \left[ (A_t - B_t K'_t)x \right] \cdot \nabla(J' - J) + \frac{1}{2}(\sigma_t \sigma_t^\dagger) \cdot \Delta(J' - J) + F(t, x) = 0,$$

where

$$F(t, x) = \left[ B_t(K_t - K'_t)x \right] \cdot \nabla J + (K'x)^\dagger R(K'x) - (Kx)^\dagger R(Kx).$$

Define $u(t, x) := J'(t, x) - J(t, x)$. By Ito's formula:

$$\mathbb{E}\left[ \mathrm{d}u(t, x'_t) \right] = \mathbb{E}\left[ F(t, x'_t) \right] \mathrm{d}t,$$

where $x'_t$ solves the state SDE (1) with $u_t$ following the policy $\pi'_t = \mathcal{N}(-K'_t x, \Sigma_t)$. Finally, by integrating on $[0, T]$, we have:

$$C(K', \Sigma) - C(K, \Sigma) = -\mathbb{E}\left[ \int_0^T \mathrm{d}u(t, x'_t) \right]$$

$$= \mathbb{E}\left[ \int_0^T F(t, x'_t)\mathrm{d}t \right].$$

A manipulation of the matrices finishes the proof. $\qquad\square$

**Lemma 10** (Contraction of IPO). *Under Assumptions 1 – 4, suppose $K'$ is the one-step update of $K$ following the algorithm* (IPO: $K$). *Then,*

$$C(K', \Sigma) - C(K^*, \Sigma) \leq \left\{ 1 - \frac{\min_{t \in [0,T]} \lambda_{\min}(y'_t)}{\max_{t \in [0,T]} ||y_t^*||_2} \right\} \left[ C(K, \Sigma) - C(K^*, \Sigma) \right],$$

*where $K^*$ is the parameter of the optimal policy, and $y'_t$ (resp. $y_t^*$) is defined by (✿).*

*Proof.* Under Assumptions 1 – 3, by Lemma 9, we have:

$$C(K, \Sigma) - C(K^*, \Sigma) = -\int_0^T \left\langle y_t^*, \mathcal{G}(t, K^*, K) \right\rangle \mathrm{d}t. \tag{19}$$

Fixing $(y_t^*)_{t \in [0,T]}$ and $(K_t)_{t \in [0,T]}$, and viewing the RHS of (19) as a functional of $(K_t^*)_{t \in [0,T]}$, we see that

$$\text{RHS of (19)} \leq \int_0^T \left\langle y_t^*, G(t, K) \right\rangle \mathrm{d}t$$

$$\leq \int_0^T ||y_t^*||_2 \operatorname{tr}\left[G(t, K)\right] \mathrm{d}t$$

$$\leq \max_{t \in [0,T]} ||y_t^*||_2 \int_0^T \operatorname{tr}\left[G(t, K)\right] \mathrm{d}t.$$

By invoking Lemma 9 again, we obtain:

$$C(K', \Sigma) - C(K, \Sigma) = -\int_0^T \left\langle y_t', G(t, K) \right\rangle \mathrm{d}t$$

$$\leq - \min_{t \in [0,T]} \lambda_{\min}(y_t') \int_0^T \operatorname{tr}\left[G(t, K)\right] \mathrm{d}t$$

$$\leq - \frac{\min_{t \in [0,T]} \lambda_{\min}(y_t')}{\max_{t \in [0,T]} ||y_t^*||_2} \left[C(K, \Sigma) - C(K^*, \Sigma)\right]. \tag{20}$$

Notice that $\max_{t \in [0,T]} ||y_t^*||_2 > 0$ by assumption 4. Adding $C(K, \Sigma) - C(K^*, \Sigma)$ to both sides of (20) gives the desired result. $\square$

**Lemma 11** (Lower bound of $\lambda_{\min}$). *Under Assumptions 1 – 4, suppose $\left\{ \left(K^{(i)}, \Sigma\right) \right\}_{i \geq 0}$ is a sequence of parameters following the algorithm* (IPO: $K$). *Then, there exists $\underline{\mu} > 0$, which is affected by $K^{(0)}$, such that:*

$$\forall i \geq 0,\, t \in [0,T], \quad \lambda_{\min}(y_t^{(i)}) \geq \underline{\mu},$$

*where $y_t^{(i)}$ is defined by ($\maltese$).*

*Proof.* Under Assumptions 1 – 3, for any fixed Gaussian policy parameterized by $(K, \Sigma)$, $y_t$ follows the ODE from Ito's formula:

$$\frac{\mathrm{d}y_t}{\mathrm{d}t} = (A_t - B_t K_t)y_t + y_t(A_t - B_t K_t)^\dagger + \sigma_t \sigma_t^\dagger, \quad y_0 = \mathbb{E}(x_0 x_0^\dagger \,|\, x_0 \sim \mathcal{D}_0).$$

Noticing that $\sigma_t \sigma_t^\dagger \succeq 0$, by adapting the proof of (Giegrich et al., 2022, Lemma 3.7), we obtain:

$$\min_{t \in [0,T]} \lambda_{\min}(y_t) \geq \lambda_{\min}(y_0) \exp\left(-2 \int_0^T ||A_t - B_t K_t||_2 \mathrm{d}t\right). \tag{21}$$

For the sequence of parameters $\left\{ \left(K^{(i)}, \Sigma\right) \right\}_{i \geq 0}$ defined by (IPO: $K$), we define $\Delta P^{(i)} := P^{K^{(i+1)}} - P^{K^{(i)}}$. It satisfies the Riccati equation (*cf.* (8)):

$$\frac{\mathrm{d}\Delta P^{(i)}}{\mathrm{d}t} + (A_t - B_t K_t^{(i+1)})^\dagger \Delta P^{(i)} + \Delta P^{(i)}(A_t - B_t K_t^{(i+1)}) = G(t, K^{(i)}), \quad \Delta P_T^{(i)} = 0.$$

Since $G(t, K^{(i)}) \succeq 0$, it implies that $\Delta P^{(i)} \preceq 0$ (*cf.* the proof of (Giegrich et al., 2022, Proposition 3.5(1))). Therefore, for any $i \geq 1$, $||K_t^{(i)}||_2 \leq ||R_t^{-1} B_t^\dagger||_2 ||P_t^{K^{(0)}}||_2$, *i.e.*, the matrix 2-norm is bounded from above. Combining this upper bound with (21) yields the desired conclusion (note that $\lambda_{\min}(y_0) > 0$ by Assumption 4). $\square$

The following lemma is immediate from Lemma 9 and Lemma 11, and by observing that:

$$\mathcal{G}(t, K, K^*) = (K_t - K_t^*)^\dagger R_t (K_t - K_t^*) \succeq 0.$$

**Lemma 12** (Upper bound of $L^2$ distance). *Under Assumptions $1-4$, suppose $\left\{\left(K^{(i)}, \Sigma\right)\right\}_{i \geq 0}$ is a sequence of parameters following the algorithm* (IPO: $K$). *Then,*

$$\forall i \geq 0, \quad C(K^{(i)}, \Sigma) - C(K^*, \Sigma) \geq \underline{\mu}\delta \int_0^T ||K_t^{(i)} - K_t^*||_2^2 \mathrm{d}t,$$

*where $\underline{\mu} > 0$ is defined in Lemma 11, and $\delta > 0$ is defined in Assumption 3.*

### 7.4 PROOF OF THEOREM 5

The proof of local super-linear convergence is built upon a series of lemmas.

**Lemma 13** (Contraction of IPO). *Under Assumptions $1-4$, suppose $\left\{\left(K^{(i)}, \Sigma\right)\right\}_{i \geq 0}$ is a sequence of parameters following the algorithm* (IPO: $K$) *and satisfying*

$$\forall i \geq 1, \quad \max_{t \in [0,T]} ||y_t^{(i)} - y_t^*||_2 \leq \min_{t \in [0,T]} \lambda_{\min}(y_t^*).$$

*Then,*

$$C(K^{(i+1)}, \Sigma) - C(K^*, \Sigma) \leq \frac{\max_{t \in [0,T]} ||y_t^{(i+1)} - y_t^*||_2}{\min_{t \in [0,T]} \lambda_{\min}(y_t^*)} \left[C(K^{(i)}, \Sigma) - C(K^*, \Sigma)\right].$$

*Proof.* Denote $K' := K^{(i+1)}$ and $K := K^{(i)}$ to simplify the notation. Under Assumptions $1-3$, by Lemma 9, we have:

$$\begin{aligned}
C(K', \Sigma) - C(K, \Sigma) &= \int_0^T \left\langle y_t', \mathcal{G}(t, K', K)\right\rangle \mathrm{d}t \\
&= -\int_0^T \left\langle y_t', G(t, K)\right\rangle \mathrm{d}t \\
&= -\int_0^T \left\langle y_t^*, G(t, K)\right\rangle - \left\langle y_t' - y_t^*, G(t, K)\right\rangle \mathrm{d}t.
\end{aligned}$$  (22)

Notice that $\min_{t \in [0,T]} \lambda_{\min}(y_t^*) > 0$ by Lemma 11 under Assumptions $1-4$. As a result,

$$\begin{aligned}
\text{RHS of (22)} &\leq \left(-1 + \frac{\max_{t \in [0,T]} ||y_t' - y_t^*||_2}{\min_{t \in [0,T]} \lambda_{\min}(y_t^*)}\right) \int_0^T \left\langle y_t^*, G(t, K)\right\rangle \mathrm{d}t \\
&\leq \left(-1 + \frac{\max_{t \in [0,T]} ||y_t' - y_t^*||_2}{\min_{t \in [0,T]} \lambda_{\min}(y_t^*)}\right) \left[C(K, \Sigma) - C(K^*, \Sigma)\right].
\end{aligned}$$

Adding $C(K, \Sigma) - C(K^*, \Sigma)$ to both sides of the above inequality gives the desired result. $\square$

**Lemma 14** (Perturbation of $y_t$). *Let $\rho > 0$. Under Assumptions $1-3$, suppose the two policies $\left\{\left(K^i, \Sigma\right)\right\}_{i=1,2}$ satisfy:*

$$\max_{1 \leq i \leq 2} \int_0^T \left||A_t - B_t K_t^i\right||_2 \mathrm{d}t \leq \rho.$$

*Then, there exists $\hat{c}_\rho > 0$ such that:*

$$\max_{t \in [0,T]} \left||y_t^1 - y_t^2\right||_2 \leq \hat{c}_\rho \int_0^T \left||K_t^1 - K_t^2\right||_2 \mathrm{d}t.$$

*Proof.* We divide the proof into several steps.

---
Step 1: Calculate the perturbation of $y_t$.
---

Under Assumptions $1-3$, by Ito's formula, $y_t$ satisfies the ODE:

$$\frac{\mathrm{d}y_t}{\mathrm{d}t} = (A_t - B_t K_t)y_t + y_t(A_t - B_t K_t)^\dagger + \sigma_t \sigma_t^\dagger, \quad y_0 = \mathbb{E}(x_0 x_0^\dagger \mid x_0 \sim \mathcal{D}_0).$$  (23)

By subtracting the ODEs that $y_t^1$ and $y_t^2$ satisfy, we get:

$$\frac{\mathrm{d}(y_t^1 - y_t^2)}{\mathrm{d}t} = (A_t - B_t K_t^1)(y_t^1 - y_t^2) + (y_t^1 - y_t^2)(A_t - B_t K_t^1)^\dagger$$

$$- \big[B_t(K_t^1 - K_t^2)\big]y_t^2 - y_t^2\big[B_t(K_t^1 - K_t^2)\big]^\dagger, \quad y_0^1 - y_0^2 = 0. \quad (24)$$

$\boxed{\textit{Step 2: Bound the norm of } y_t.}$

By integrating over $[0, t]$ and then taking norms on both sides of (23), we get:

$$||y_t||_2 \leq ||y_0||_2 + 2\int_0^t ||A_s - B_s K_s||_2 ||y_s||_2 + ||\sigma_s \sigma_s^\dagger||_2 \mathrm{d}s.$$

By Gronwall's inequality, there exists $\tilde{c}_\rho > 0$ such that:

$$\max_{t \in [0,T]} ||y_t||_2 \leq \tilde{c}_\rho.$$

$\boxed{\textit{Step 3: Bound the perturbation of } y_t.}$

By integrating over $[0, t]$ and then taking norms on both sides of (24), we get:

$$||y_t^1 - y_t^2||_2 \leq 2\int_0^t ||A_s - B_s K_s^1||_2 ||y_s^1 - y_s^2||_2 + ||B_s||_2 ||K_s^1 - K_s^2||_2 ||y_s^2||_2 \mathrm{d}s$$

$$\leq 2\int_0^t ||A_s - B_s K_s^1||_2 ||y_s^1 - y_s^2||_2 \mathrm{d}s + 2\tilde{c}_\rho \max_{t \in [0,T]} ||B_t||_2 \int_0^t ||K_s^1 - K_s^2||_2 \mathrm{d}s.$$

Again, by Gronwall's inequality, there exists $\hat{c}_\rho > 0$ such that:

$$\max_{t \in [0,T]} ||y_t^1 - y_t^2||_2 \leq \hat{c}_\rho \int_0^T ||K_t^1 - K_t^2||_2 \mathrm{d}t.$$

$\square$

**Lemma 15** (Bound the one-step update of $y_t$). *Under Assumptions 1 – 3, let $\rho > 0$ be such that*

$$\int_0^T ||A_t - B_t K_t^*||_2 \mathrm{d}t \leq \rho.$$

*Suppose $\big\{\big(K^{(i)}, \Sigma\big)\big\}_{i \geq 0}$ is a sequence of parameters following the algorithm* (IPO: $K$) *and satisfying:*

$$\sup_{i \geq 0} \int_0^T ||A_t - B_t K_t^{(i)}||_2 \mathrm{d}t \leq \rho.$$

*Then, there exists $c_\rho^* > 0$ which is affected by $K^{(0)}$, such that for any $i \geq 0$, we have:*

$$\forall i \geq 0, \quad \max_{t \in [0,T]} ||y_t^{(i+1)} - y_t^*||_2 \leq c_\rho^* \int_0^T ||K_t^{(i)} - K_t^*||_2 \mathrm{d}t.$$

*Proof.* Denote $K' := K^{(i+1)}$ and $K := K^{(i)}$. By definition,

$$||K_t' - K_t^*||_2 = ||R_t^{-1} B_t^\dagger (P_t^K - P_t^{K^*})||_2$$

$$\leq \frac{||B_t||_2}{\lambda_{\min}(R_t)} ||P_t^K - P_t^{K^*}||_2,$$

where $\lambda_{\min}(R_t) \geq \delta > 0$ for any $t \in [0, T]$ by Assumption 3.

Notice that $P_t^K - P_t^{K^*}$ satisfies the ODE:

$$\frac{\mathrm{d}(P_t^K - P_t^{K^*})}{\mathrm{d}t} = (A_t - B_t K_t)^\dagger (P_t^K - P_t^{K^*}) + (P_t^K - P_t^{K^*})(A_t - B_t K_t) + K_t^\dagger R_t K_t$$

$$- \big[B_t(K_t - K_t^*)\big]^\dagger P_t^{K^*} - P_t^{K^*}\big[B_t(K_t - K_t^*)\big] - (K_t^*)^\dagger R_t K_t^*, \quad P_T^K - P_T^{K^*} = 0.$$

By integrating over $[t, T]$ and taking norms on both sides, we get:

$$||P_t^K - P_t^{K^*}||_2 \leq 2 \int_t^T ||A_s - B_s K_s||_2 ||P_s^K - P_s^{K^*}||_2 \mathrm{d}s$$

$$+ 2 \max_{s \in [0,T]} ||B_s^\dagger P_s^{K^*}||_2 \int_t^T ||K_s - K_s^*||_2 \mathrm{d}s$$

$$+ \max_{s \in [0,T]} ||R_s||_2 \int_t^T \left( ||K_s||_2 + ||K_s^*||_2 \right) ||K_s - K_s^*||_2 \mathrm{d}s.$$

Recall from the proof of Lemma 11 that $\left\| K_s \right\|_2 \leq \left\| R_s^{-1} B_s^\dagger \right\|_2 \left\| P_s^{K^{(0)}} \right\|_2$, which only requires Assumptions $1 - 3$. As a result,

$$||P_t^K - P_t^{K^*}||_2 \leq 2 \int_t^T ||A_s - B_s K_s||_2 ||P_s^K - P_s^{K^*}||_2 \mathrm{d}s$$

$$+ \left[ 2 \max_{s \in [0,T]} ||B_s^\dagger P_s^{K^*}||_2 + \max_{s \in [0,T]} ||R_s||_2 \left( \max_{s \in [0,T]} ||R_s^{-1} B_s^\dagger||_2 ||P_s^{K^{(0)}}||_2 \right. \right.$$

$$\left. \left. + \max_{s \in [0,T]} ||K_s^*||_2 \right) \right] \int_t^T ||K_s - K_s^*||_2 \mathrm{d}s.$$

Therefore, by Gronwall's inequality, there exists $\bar{c}_\rho > 0$, which is affected by $K^{(0)}$, such that:

$$\max_{t \in [0,T]} ||P_t^K - P_t^{K^*}||_2 \leq \bar{c}_\rho \int_0^T ||K_t - K_t^*||_2 \mathrm{d}t,$$

and moreover,

$$\max_{t \in [0,T]} ||K' - K_t^*||_2 \leq \bar{c}_\rho \max_{t \in [0,T]} \frac{||B_t||_2}{\lambda_{\min}(R_t)} \int_0^T ||K_t - K_t^*||_2 \mathrm{d}t.$$

Finally, noticing that $\int_0^T ||A_t - B_t K_t^*||_2 \mathrm{d}t \leq \rho$, an application of Lemma 14 finishes the proof. $\quad\square$

*Proof of Theorem 5.* To show the existence of $\epsilon$, denote $r := \int_0^T ||K_t^{(0)} - K_t^*||_2^2 \mathrm{d}t$. Recall from the proof of Lemma 11 that $||K_t^{(i)}||_2 \leq ||R_t^{-1} B_t^\dagger||_2 ||P_t^{K^{(0)}}||_2$ for any $i \geq 1$, which only requires Assumptions $1 - 3$. By applying Gronwall's inequality on (8), $\max_{t \in [0,T]} ||P_t^{K^{(0)}}||_2$ is bounded from above, and the bound only depends on the value of $r$ (as an increasing function in $r$) and the data of the LQR. As a result, there exists $\rho_r > 0$, which only depends on the value of $r$ (as an increasing function in $r$) and the data of the LQR, such that

$$\max \left\{ \int_0^T ||A_t - B_t K_t^*||_2 \mathrm{d}t, \sup_{i \geq 0} \int_0^T ||A_t - B_t K_t^{(i)}||_2 \mathrm{d}t \right\} \leq \rho_r.$$

Under Assumptions $1 - 4$, by Lemma 11 and Lemma 12, there exists $\underline{\mu}_r > 0$, which only depends on the value of $r$ (as a decreasing function in $r$) and the data of the LQR, such that

$$\forall i \geq 0, \quad \underline{\mu}_r \delta \int_0^T ||K_t^{(i)} - K_t^*||_2^2 \mathrm{d}t \leq C(K^{(i)}, \Sigma) - C(K^*, \Sigma).$$

Meanwhile, by applying Gronwall's inequality on (23), there exists $\bar{\mu}_r > 0$, which only depends on the value of $r$ (as an increasing function in $r$) and the data of the LQR, such that

$$\forall i \geq 0, \quad \max_{t \in [0,T]} ||y_t^{(i)}||_2 \leq \bar{\mu}_r.$$

As a result, by Lemma 9,

$$\forall i \geq 0, \quad C(K^{(i)}, \Sigma) - C(K^*, \Sigma) \leq \bar{\mu}_r \max_{t \in [0,T]} ||R_t||_2 \int_0^T ||K_t^{(i)} - K_t^*||_2^2 \mathrm{d}t.$$

By Lemma 15, there exists $c_r^* > 0$, which only depends on the value of $r$ (as an increasing function in $r$) and the data of the LQR, such that:

$$\forall i \geq 0, \quad \max_{t \in [0,T]} ||y_t^{(i+1)} - y_t^*||_2 \leq c_r^* \int_0^T ||K_t^{(i)} - K_t^*||_2 \mathrm{d}t.$$

Putting everything together, for any $i \geq 0$, we have:

$$\max_{t \in [0,T]} ||y_t^{(i+1)} - y_t^*||_2 \leq c_r^* \int_0^T ||K_t^{(i)} - K_t^*||_2 \mathrm{d}t$$

$$\leq c_r^* \sqrt{T} \sqrt{\int_0^T ||K_t^{(i)} - K_t^*||_2^2 \mathrm{d}t}$$

$$\leq c_r^* \sqrt{\frac{T}{\underline{\mu}_r \delta}} \sqrt{C(K^{(0)}, \Sigma) - C(K^*, \Sigma)}$$

$$\leq c_r^* \sqrt{\frac{T \bar{\mu}_r \max_{t \in [0,T]} ||R_t||_2}{\underline{\mu}_r \delta}} \sqrt{r}, \tag{25}$$

where $\delta > 0$ is defined in Assumption 3. Since the RHS of (25) is an increasing function in $r$ and tends to 0 as $r \to 0^+$, there exists $\epsilon > 0$, such that $r < \epsilon$ implies

$$\forall i \geq 1, \quad \max_{t \in [0,T]} ||y_t^{(i)} - y_t^*||_2 \leq \min_{t \in [0,T]} \lambda_{\min}(y_t^*).$$

This proves the existence of $\epsilon$.

Finally, to calculate the corresponding $\mathcal{C}_2$, by Lemma 13,

$$\forall i \geq 0, \quad C(K^{(i+1)}, \Sigma) - C(K^*, \Sigma)$$

$$\leq \frac{\max_{t \in [0,T]} ||y_t^{(i+1)} - y_t^*||_2}{\min_{t \in [0,T]} \lambda_{\min}(y_t^*)} \left[ C(K^{(i)}, \Sigma) - C(K^*, \Sigma) \right]$$

$$\leq \frac{c_r^* \sqrt{T}}{\min_{t \in [0,T]} \lambda_{\min}(y_t^*)} \sqrt{\int_0^T ||K_t^{(i)} - K_t^*||_2^2 \mathrm{d}t} \left[ C(K^{(i)}, \Sigma) - C(K^*, \Sigma) \right]$$

$$\leq \frac{c_r^*}{\min_{t \in [0,T]} \lambda_{\min}(y_t^*)} \sqrt{\frac{T}{\underline{\mu}_r \delta}} \left[ C(K^{(i)}, \Sigma) - C(K^*, \Sigma) \right]^{\frac{3}{2}},$$

*i.e.*, $\mathcal{C}_2 = \frac{c_r^*}{\min_{t \in [0,T]} \lambda_{\min}(y_t^*)} \sqrt{\frac{T}{\underline{\mu}_r \delta}}.$ $\qquad\square$

