# OpenReview forum: "Policy transfer ensures fast learning for continuous-time LQR with entropy regularization"
_ICLR.cc/2026/Conference — Submitted to ICLR 2026_

### Official Review · Reviewer_UBHz · 2025-10-31

**Soundness:** 3
**Presentation:** 2
**Contribution:** 3
**Rating:** 4
**Confidence:** 3

**Summary:**

This paper presents the first theoretical framework for policy transfer in continuous-time RL, focusing on entropy-regularized LQRs. The authors rigorously prove that an optimal policy for one LQR remains near-optimal for a closely related LQR, thereby ensuring efficient transfer and fast convergence.

**Strengths:**

This is the first formal proof of policy transferability in continuous-time RL, a problem significantly harder than its discrete counterpart. The results establish a solid foundation for understanding transfer efficiency under system perturbations. The proofs are rigorous. By generalizing the discrete-time IPO results to continuous-time, the paper fills a conceptual and technical gap between theory and control-oriented RL applications.

**Weaknesses:**

The paper is purely theoretical, lacking numerical simulations or empirical studies that could illustrate the benefits or limitations of policy transfer in practice.

Moreover, the notion of “closeness” between two LQRs is not concretely quantified. All results are stated in the general form of “as long as they are sufficiently close,” which limits the practical applicability of the theory for algorithm design.

**Questions:**

1. Ambiguity regarding the notion of "closeness":
    - In Theorem 1, it is unclear how large one should expect $\epsilon$ to be, especially under specific choices of the metric $d$?
    - In Theorem 5, similar clarification is needed on the expected magnitudes of $\epsilon$ and $C_2$?
2. It would be helpful if the authors could provide a simple numerical experiment to illustrate or verify the theoretical results.
3. Assumptions 4-5 are presented without any accompanying discussion. It would be valuable if the authors could elaborate on how restrictive these assumptions are in practice.
4. Throughout the proofs, Assumptions 1–5 are invoked but not explicitly referenced when applied. Clearer indication of where each assumption is used would improve the readability and rigor of the arguments.

---

> ### Author Response · Authors · 2025-11-25
>
> Thank you for the review.
>
> For the first question, since the proof of Theorem 1 is based on Lemma 2 and Lemma 3, the choice of $d$ can be found in Lemma 3 and $\epsilon$ comes from the definition of continuity (i.e., the so-called $\epsilon$-$\delta$ language). In Theorem 5, the values of $\epsilon$ and $\mathcal{C}_2$ can be found in Section 7.4 (i.e., the proof of Theorem 5). It is hard to write out their values completely since we used Gronwall's inequality in multiple  places of the proof.
>
> For the second question, we have conducted a toy numerical experiment, and the results have been added in the newly added section "Numerical Experiments."  The error curve clearly indicated that our proposed IPO algorithm admits global linear convergence and local super-linear convergence.
>
> For the third question, note that we discussed Assumption 4 in Remark 2 and Remark 3. We apologize that we did not discuss Assumption 5 in our first version PDF. In fact, we do not think Assumption 5 will pose many restrictions in practice. To see this, suppose given an arbitrary diffusion model in the form of Eqn. (13) (i.e., the SDE is an OU process and the data distribution is Gaussian). Then, one can always construct an LQR such that Lemma 7 holds. For example, one may first choose $k = d$ and define $R_t$ as a rescaled identity matrix according to the first equality in Assumption 5. Second, one may define $B_t$ as a rescale of $\sigma_t$ according to the second equality in Assumption 5. The rest of parameters of the LQR can be defined easily.
>
> For the fourth question, we apologize if there was any confusion, as  we already had the references of Assumptions 1-5 in the statement of the theorems and their proofs. We are happy to provide a further clarification if you can kindly let us know specifically where you think the unclarity is.

---

> > ### Comment · Reviewer_UBHz · 2025-11-26
> >
> > I thank the authors for their responses. Regarding question 1, my concern is that the theorem does not specify the order of closeness: given a set of problem parameters, how large can $\epsilon$ be? Without such characterization, the theorem does not provide actionable guidance on when provable transfer can be achieved. For example, in the episodic RL setting, if the total variation distance between two task is $O(\varepsilon/H^2)$, then the optimal policy for one task is $O(\varepsilon)$-optimal for the other task [1]. Can the authors provide an analogous quantitative relationship in your setting?
> >
> > For question 2 and 3, thanks for the further clarification.
> >
> > For question 4, in the proofs of Lemmas 9–15, the assumptions are not explicitly referenced at the points where they are applied; they are stated only in the lemma statements.
> >
> > > [1] An Optimal Tightness Bound for the Simulation Lemma. arXiv 2024.

---

> > > ### Author Response · Authors · 2025-11-26
> > >
> > > Thank you for the reply.
> > >
> > > For question 1, it is hard to give a quantitative relationship due to Gronwall's inequality.
> > >
> > > For question 4, this is a good point and we have added more explanations in the lemma proofs. Please let us know if there is still any unclarity.

---

### Official Review · Reviewer_W6Gx · 2025-10-31

**Soundness:** 2
**Presentation:** 2
**Contribution:** 2
**Rating:** 4
**Confidence:** 5

**Summary:**

Summary: The paper studies policy transfer in reinforcement learning (RL) for continuous-time linear quadratic regulators (LQRs) with entropy regularization. The paper proves that the optimal policy of a source LQR task provides a near-optimal initialization for a target LQR with similar system parameters. The main contributions include: (i) a continuity result for the Riccati operator guaranteeing policy transferability between closely related LQRs; (ii) an Iterative Policy Optimization (IPO) algorithm with provable global linear and local super-linear convergence; and (iii) an application of the analysis to the stability of score-based diffusion models via the connection between LQRs and controlled SDEs.

**Strengths:**

1) The paper is well-written and well-organized. It provides an approach to connect entropy-regularized continuous-time LQRs with transfer learning. The proofs are sound and build upon solid stochastic control foundations (e.g., Riccati continuity, DPP, HJB).

2) The connection between LQRs and score-based diffusion models is interesting and well-motivated, illustrating broader implications of LQR analysis beyond control theory.

**Weaknesses:**

1) The paper frames policy transfer as a transfer learning problem but overlooks the extensive literature on meta-learning for control and meta-LQR. Examples:

[1] Toso et al. "Meta-learning linear quadratic regulators: a policy gradient MAML approach for model-free LQR." 6th Annual Learning for Dynamics \& Control Conference. PMLR, 2024.

 [2] Aravind et al. "A Moreau envelope approach for LQR meta-policy estimation." 2024 IEEE 63rd Conference on Decision and Control (CDC). IEEE, 2024.

 [3] Muthirayan et al. "Meta-learning online control for linear dynamical systems." IEEE Transactions on Automatic Control (2025).

 [4] Richards et al. "Control-oriented meta-learning." The International Journal of Robotics Research (2023)

 [5] Zhang et al. "Multi-task imitation learning for linear dynamical systems." Learning for Dynamics and Control Conference. PMLR, 2023.

2) The aforementioned work and follow-up analyses on meta-adaptation and task similarity are highly relevant, as they provide theoretical and algorithmic frameworks for policy reuse across task distribution, precisely the problem studied here. Without comparing with respect to this line of work, the contribution risks appearing as an isolated instance of LQR transfer rather than a step toward data-efficient multi-task or meta-RL.

3) The IPO algorithm seems structurally identical to the discrete-time IPO in Guo et al. (2023), with the main novelty being a continuous-time adaptation. No significant methodological innovation or empirical verification seems to be introduced.

4) Although the work is primarily theoretical, even simple numerical examples (e.g., two LQRs with slightly perturbed dynamics) would enhance clarity regarding the policy transfer effect and convergence rates.

5) The exposition of transfer learning motivations is extensive and heavily references LLM/vision TL literature, but the discussion of control-specific challenges (e.g., transfer between dynamical systems, partial observability) is brief and lacks comparison to established control-theoretic notions of similarity or adaptation.

**Questions:**

1) How does the proposed framework compare to recent meta-LQR formulations, where task similarity is quantified via parameter-space or trajectory-space distances?

2) Could the authors formalize the notion of  "closely related" LQRs using information-theoretic or bisimulation-based measures, as often done in meta-control literature?

3) How might the entropy regularization interact with multi-task adaptation or robustness to heterogeneity across tasks?

Comments:

1) The exposition is clear and mathematically precise, though the paper could better contextualize its novelty relative to multi-task and meta-RL control frameworks.

2) The link to diffusion models is interesting but somewhat tangential; emphasizing implications for meta-control or representation transfer might make the contribution more cohesive.

---

> ### Author Response · Authors · 2025-11-25
>
> Thank you for the review.
>
> For the first question, [1] proves the stability and convergence results of the model-agnostic meta-learning (MAML) in discrete-time LQRs. Our work in contrast is on the continuous-time  where the distance between continuous-time LQRs is measured by parameter-space distances,  and the technical difficulty comes from dealing with the infinite dimensional functional spaces. For instance, one needs to analyze the stability of the associated Ricatti equation. Note that the parameters in our setting are continuous-time trajectories. We added [1] to the "Related work" section.
>
> For the second question, it was not completely clear to us and please can you be more specific.
>
> For the third question, entropy regularization encourages agent exploration in general. However, in our work, the definition of the IPO algorithm is independent of the entropy regularization term. This means entropy regularization is not crucial for our analysis (e.g., consider the special case where $\tau = 0$).
>
> Reference:
>
> [1] Toso, Leonardo Felipe, et al. "Meta-learning linear quadratic regulators: a policy gradient MAML approach for model-free LQR." 6th Annual Learning for Dynamics & Control Conference. PMLR, 2024.

---

> > ### Comment · Reviewer_W6Gx · 2025-11-25
> >
> > I thank the authors for the response. What I meant with question 2 is for the authors to provide some clarifications in the notion of closeness between LQR tasks discussed in this work and make connections with heterogeneity measures in the literature. Could it also be characterized through information-theoretic or bisimulation-based measures as often discussed multitask learning for control? A related question would be: what is the effect of heterogeneity in the learning and whether the way heterogeneity measure discussed in the paper is tight/conservative compared to closed-loop heterogeneity measures.

---

> > > ### Author Response · Authors · 2025-11-28
> > >
> > > Thank you for the reply.
> > >
> > > In our work, we view $(A_{[0, T]}, Q_{[0, T]}, B_{[0, T]}, R_{[0, T]}, Q^\prime)$ as the parameters of an LQR. And the distance between two LQRs is defined by the distance between their parameters. Note that here $(A_{[0, T]}, Q_{[0, T]}, B_{[0, T]}, R_{[0, T]})$ are continuous-time trajectories, and $Q^\prime$ is a matrix.
> > >
> > > We believe that it is possible to translate our distance between two LQRs into the language of heterogeneity measure in meta learning. For example, suppose we have a group of LQRs. The (empirical) distribution of them can be represented by the (empirical) distribution of their parameters which take value in infinite-dimensional spaces.
> > >
> > > We also believe that our distance can be characterized by bisimulation-based measures. The idea is intuitive: two closely related LQRs (i.e., the distance between them is small) should have almost the same state dynamic and cost function. By "almost" we mean the distance (which is chosen properly) between their dynamics and cost functions is small.
> > >
> > > Finally, in our work, we showed that the parameters of the optimal Gaussian policy of an LQR are continuous in the LQR's parameters. As a result, if the distance between two LQRs is small, then the parameters of their optimal Gaussian policies will also be close to each other. From another point of view, since Gaussian distribution is continuous in its mean and covariance (the probability space is equipped with the weak topology, for example), the optimal policy of an LQR, when viewed as a flow of probability measures, is also continuous in the LQR's parameters.

---

### Official Review · Reviewer_fn1S · 2025-11-02

**Soundness:** 3
**Presentation:** 3
**Contribution:** 1
**Rating:** 4
**Confidence:** 4

**Summary:**

This paper investigates policy transfer for continuous-time linear quadratic regulators (LQRs) with entropy regularization, extending prior analyses from discrete-time to continuous-time settings. The authors introduced an Iterative Policy Optimization (IPO) algorithm that achieves global linear and local super-linear convergence. As a byproduct of their analysis, they establish the stability of a class of continuous-time score-based diffusion models through their connection with LQRs.

**Strengths:**

1. The theoretical results are supported by clean mathematical derivations, leveraging properties of the Riccati equation and Gaussian policy structures.

2. The results provide a formal justification for policy reuse and initialization transfer between related LQR systems.

3. The connection between LQRs and score-based diffusion models is interesting, yielding new stability results with potential implications for generative modeling.

**Weaknesses:**

1. The theoretical development appears to be a relatively direct extension of the prior work [1], without substantial methodological novelty. If the authors believe that extending the results to continuous settings is non-trivial, the paper should clearly articulate the specific challenges unique to the continuous-time setting and how their approach overcomes them.

2. The proposed IPO algorithm also appears to be a direct extension of the discrete-time version in [1]. The authors could strengthen the paper by clarifying what is fundamentally new in the continuous-time formulation, whether in algorithmic design, analysis, or convergence guarantees.


[1] Guo, Xin, Xinyu Li, and Renyuan Xu. "Fast policy learning for linear quadratic control with entropy regularization." arXiv preprint arXiv:2311.14168 (2023).

**Questions:**

1. It would strengthen the paper to discuss theoretical works on policy transfer in MDP settings, such as [2].

2. While the theoretical contributions are clear, the paper would benefit from empirical validation to illustrate the practical relevance of the results. Even simple numerical experiments on continuous-time LQRs could help demonstrate the effectiveness and convergence behavior of the proposed IPO algorithm and the benefits of policy transfer in practice.


[2] Fu, Haotian, et al. "Performance bounds for model and policy transfer in hidden-parameter mdps." The Eleventh International Conference on Learning Representations. 2023.

---

> ### Author Response · Authors · 2025-11-25
>
> Thank you for the review.
>
> For the first question, we added a paragraph in the "Related work" section about transfer learning (TL) between MDPs. There has been much work on this topic, e.g., [1], [2], and [3],  to name a few. [1] investigates model transfer and policy transfer between hidden-parameter MDPs (HiP-MDPs), bounding the performance loss incurred by TL with the error in the estimation of hidden parameters. [2] proposes sample-transfer algorithms and conducts the corresponding finite-sample analysis. [3] proves that, within the class of Lipschitz continuous MDPs, small perturbations in the dynamics only lead to a small change in the value function.
>
> For the second question, we conducted a toy numerical experiment, and the results were presented in the newly added section "Numerical Experiments."  The error curve clearly indicated that our proposed IPO algorithm admits global linear convergence and local super-linear convergence.
>
> References:
>
> [1] Fu, Haotian, et al. "Performance bounds for model and policy transfer in hidden-parameter mdps." The Eleventh International Conference on Learning Representations. 2023.
>
> [2] Lazaric, Alessandro, and Marcello Restelli. "Transfer from multiple MDPs." Advances in neural information processing systems 24 (2011).
>
> [3] Asadi, Kavosh, Dipendra Misra, and Michael Littman. "Lipschitz continuity in model-based reinforcement learning." International conference on machine learning. PMLR, 2018.

---

### Meta-Review · Area_Chair_EdUB · 2026-01-10

**Summary:**

All three reviewers rated the paper as "marginally below the acceptance threshold". Their primary concerns, which informed the decision to reject, are as follows:

Reviewers 1 and 2 noted that the theoretical development and the proposed algorithm appear to be a relatively direct extension of prior discrete-time work (Guo et al., 2023) to the continuous-time setting. They found the paper did not sufficiently articulate the unique challenges of the continuous-time formulation or demonstrate substantial methodological innovation.

Reviewers 1 also highlighted that the paper overlooks or insufficiently discusses closely related work, particularly in meta-learning for control and policy transfer in MDPs.

While acknowledging the paper's theoretical nature, all three reviewers initially noted the absence of numerical experiments to illustrate the proposed algorithm.

**Reviewer Concerns:**

Addressed Concerns:

The authors added a numerical experiment in the rebuttal, which partially addresses the request for empirical validation.

The authors expanded the related work section to include references on transfer learning in MDPs (addressing Reviewer 1) and meta-LQR (addressing Reviewer 2).

Outstanding Concerns:

Core novelty remains disputed; the continuous-time extension and IPO algorithm are still perceived as direct adaptations.

The key request for a quantitative, interpretable bound on task similarity was not met.

Concerns about restrictive assumptions and proof tightness were not fully resolved.

**Reviewer Scores:**

Reviewer fn1S: Initial score 4. The added experiment and related work are positive but do not address the core novelty concern. The score would likely remain at 4.

Reviewer W6Gx: Initial score 4. The lack of a compelling response on methodological novelty suggests the score might remain at 4.

Reviewer UBHz: Initial score 4. The numerical experiment is appreciated, but the failure to provide a quantitative characterization of "closeness" is a significant shortcoming. The score might remain at 4.

---

### Decision · Program_Chairs · 2026-01-26

Reject